# Strong Effective Coupling, Meson Ground States, and Glueball within Analytic Confinement

**Gurjav Ganbold** [1,2]

1    Bogoliubov Laboratory of Theoretical Physics, JINR, Joliot-Curie 6, 141980 Dubna, Russia; ganbold@theor.jinr.ru
2    Institute of Physics and Technology, Mongolian Academy of Sciences, Enkh Taivan 54b,
     13330 Ulaanbaatar, Mongolia

**Abstract:** The phenomena of strong running coupling and hadron mass generating have been studied in the framework of a QCD-inspired relativistic model of quark-gluon interaction with infrared-confined propagators. We derived a meson mass equation and revealed a specific new behavior of the mass-dependent strong coupling $\hat{\alpha}_s(M)$ defined in the time-like region. A new infrared freezing point $\hat{\alpha}_s(0) = 1.03198$ at origin has been found and it did not depend on the confinement scale $\Lambda > 0$. Independent and new estimates on the scalar glueball mass, 'radius' and gluon condensate value have been performed. The spectrum of conventional mesons have been calculated by introducing a minimal set of parameters: the masses of constituent quarks and $\Lambda$. The obtained values are in good agreement with the latest experimental data with relative errors less than 1.8 percent. Accurate estimates of the leptonic decay constants of pseudoscalar and vector mesons have been performed.

**Keywords:** quark model; confinement; strong coupling; meson; glueball; leptonic decay

## 1. Introduction

The low-energy region below ∼2 GeV becomes a testing ground, where much novel, interesting and challenging behavior is revealed in particle physics (see, e.g., [1]). Any QCD-inspired theoretical model should be able to correctly describe hadron phenomena such as confinement, running coupling, hadronization, mass generation etc. at large distances. The inefficiency of the conventional perturbation theory in low-energy domain pushes particle physicists to develop and use different phenomenological and nonperturbative approaches, such as QCD sum rule, chiral perturbation theory, heavy quark effective theory, rigorous lattice QCD simulations, the coupled Schwinger-Dyson equation etc.

The confinement conception explaining the non-observation of color charged particles (quarks, gluons) is a crucial feature of QCD and a great number of theoretical models have been suggested to explain the origin of confinement. Particularly, the confinement may be parameterized by introducing entire-analytic and asymptotically free propagators [2]), vacuum gluon fields serving as the true minimum of the QCD effective potential [3], self-dual vacuum gluon fields leading to the confined propagators [4], the Wilson loop techniques [5], lattice Monte-Carlo simulations [6], a string theory quantized in higher dimensions [7] etc. Each approach has its benefits, justifications, and limitations. A simple and reliable working tool implementing the confinement concept is still required.

The strength of quark-gluon interaction $g$ in QCD depends on the mass scale or momentum transfer $Q$. This dependence is described theoretically by the renormalization group equations and the behavior of $\alpha_s \doteq g^2/(4\pi)$ at short distances (for high $Q^2$), where asymptotic freedom appears, is well investigated

[8,9] and measured, e.g., $\alpha_s(M_Z^2) = 0.1185 \pm 0.0006$ at mass scale $M_Z = 91.19$ GeV [1]. On the other hand, it is necessary to know the long-distance (for $Q^2 \leq 1$ GeV) or, infrared (IR),behavior of $\alpha_s$ to understand quark confinement, hadronization processes, and hadronic structure. Many phenomena in particle physics are affected by the long-distance property of the strong coupling [9,10], however the IR behavior of $\alpha_s$ has not been well defined yet, it needs to be more specified. A self-consistent and physically meaningful prediction of $\alpha_s$ in the IR region is necessary.

The existence of extra isoscalar mesons is predicted by QCD and in case of the pure gauge theory they contain only gluons, and are called the glueballs, the bound states of gluons. Presently, glueballs are the most unusual particles predicted by theory, but not found experimentally yet [1,11]. The study of glueballs currently is performed either within effective models or lattice QCD. The glueball spectrum has been studied by using the QCD sum rules [12], Coulomb gauge QCD [13], various potential models [14]. A proper inclusion of the helicity degrees of freedom can improve the compatibility between lattice QCD and potential models [15]. Recent lattice calculations, QCD sum rules, 'tube' and constituent 'glue' models predict that the lightest glueball takes the quantum numbers ($J^{PC} = 0^{++}$) [16]. However, errors on the mass predictions are large, particularly, $M_G = 1750 \pm 50 \pm 80$ MeV for the mass of scalar glueball from quenched QCD [17]. Therefore, an accurate prediction of the glueball mass combined with other reasonable unquenched estimates and performed within a theoretical model with fixed global parameters is important.

One of the puzzles of hadron physics is the origin of the hadron masses. The Standard Model and, in particular, QCD operate only with fundamental particles (quarks, leptons, neutrinos), gauge bosons and the Higgs. It is not yet clear how to explain the appearance of the multitude of observed hadrons and elucidate the generation of their masses. Physicists have proposed several models that advocate different mechanism of the origin of mass from the most fundamental laws of physics. Particularly, the dynamical chiral symmetry breaking is one of the widely accepted mechanisms explaining the connection between the 'current' quark masses and the 'constituent' masses of quarks inside hadrons (see, e.g., [18]). Nevertheless, the calculation of the hadron mass spectrum in a quality comparable to the precision of experimental data remains actual.

In some cases, it is useful to investigate the corresponding low-energy effective theories instead of tackling the fundamental theory itself. Indeed, data interpretations and calculations of hadron characteristics are frequently carried out with the help of phenomenological models.

One of the phenomenological approaches is the model of induced quark currents. It is based on the hypothesis that the QCD vacuum is realized by the anti-selfdual homogeneous gluon field [19]. The confining properties of the vacuum field and chiral symmetry breaking can explain the distinctive qualitative features of the meson spectrum: mass splitting between pseudoscalar and vector mesons, Regge trajectories, and the asymptotic mass formulas in the heavy quark limit. Numerically, this model describes to within ten percent accuracy the masses and weak decay constants of mesons.

A relativistic constituent quark model developed first in [20] has found numerous applications both in the meson sector (e.g., [21]) and in baryon physics (e.g., [22]). In the latter case baryons are considered as relativistic systems composed of three quarks. The next step in the development of the model has been done in [23], where infrared confinement for a quark-antiquark loop was introduced. The implementation of quark confinement allowed to use the same values for the constituent quark masses both for the simplest quark-antiquark systems (mesons) and more complicated multiquark configurations (baryons, tetraquarks, etc.). Recently, a smooth decreasing behavior of the Fermi coupling on mass scale has been revealed by considering meson spectrum within this model [24].

In a series of papers [25–29] relativistic models with specific forms of analytically confined propagators have been developed to study different aspects of low-energy hadron physics. Particularly, the role of analytic confinement in the formation of two-particle bound states has been analyzed within a simple

Yukawa model of two interacting scalar fields, the prototypes of 'quarks' and 'gluons'. The spectra of the 'two-quark' and 'two-gluon' bound states have been defined by using master constraints similar to the ladder Bethe-Salpeter equations. The 'scalar confinement' model could explain the asymptotically linear Regge trajectories of 'mesonic' excitations and the existence of massive 'glueball' states [25]. An extension of this model has been provided by introducing color and spin degrees of freedom, different masses of constituent quarks and the confinement size parameter that resulted in an estimation of the meson mass spectrum (with relative errors <3.5 per cent) in a wide energy range [27]. As a further test, the weak decay constants of light mesons and the lowest-state glueball mass has been estimated with reasonable accuracies. Then, a phenomenological model with specific forms of infrared-confined propagators has been developed to study the mass-scale dependence of the QCD effective coupling $\alpha_s$ at large distances [28,29]. By fitting the physical masses of intermediate and heavy mesons we predicted a new behavior of $\alpha_s(M)$ in the low-energy domain, including a new, specific and finite behavior of $\alpha_s(M)$ at origin. Note, $\alpha_s(0)$ depended on $\Lambda$, we fixed $\alpha_s(0) = 0.757$ for $\Lambda = 345$ MeV in [28].

In the present paper, we propose a new insight into the phenomena of strong running coupling and hadron mass generating by introducing infrared-confined propagators within a QCD-inspired relativistic field model. First, we derive a meson mass master equation similar to the ladder Bethe-Salpeter equation and study a specific new behavior of the mass-dependent strong coupling $\hat{\alpha}_s(M)$ in the time-like region. Then, we estimate properties of the lowest-state glueball, namely its mass and 'radius'. The spectrum of conventional mesons are estimated by introducing a minimal set of parameters. An accurate estimation of the leptonic decay constants of pseudoscalar and vector mesons is also performed.

The paper is organized as follows. After the introduction, in Section 2 we give a brief sketch of main structure and specific features of the model, including the ultraviolet regularization of field and strong charge as well as the infrared regularizations of the propagators in the confinement domain. A self-consistent mass-dependent effective strong coupling is derived and investigated in Section 3. The formation of an exotic di-gluon bound state, the glueball, its ground-state properties are considered in Section 4. Hereby we fix the global parameter of our model, the confinement scale $\Lambda = 236$ MeV. In Sections 5 and 6 we give the details of the calculations for the mass spectrum and leptonic (weak) decay constants of the ground-state mesons in a wide range of scale. Finally, in Section 7 we summarize our findings.

## 2. Model

Consider the gauge invariant QCD Lagrangian:

$$\mathcal{L} = -\frac{1}{4}\left(\partial^\mu \mathcal{A}_\nu^A - \partial^\nu \mathcal{A}_\mu^A - gf^{ABC}\mathcal{A}_\mu^B \mathcal{A}_\nu^C\right)^2 + \left(\bar{q}_f^a \left[\gamma_\alpha \partial^\alpha - m_f\right]^{ab} q_f^b\right) + g\left(\bar{q}_f^a \left[\Gamma_C^\alpha \mathcal{A}_\alpha^C\right]^{ab} q_f^b\right), \quad (1)$$

where $\mathcal{A}_\alpha^C$ is the gluon field, $q_f^a$ is a quark spinor of flavor $f$ with color $a = \{1, 2, 3\}$ and mass $m_f = \{m_{ud}, m_s, m_c, m_b\}$, $\Gamma_C^\alpha = i\gamma_\alpha t^C$ and $g$ - the strong coupling strength.

Below we study two-particle bound state properties within the model. The leading-order contributions to the spectra of quark-antiquark and di-gluon bound states are given by the partition functions:

$$Z_{q\bar{q}} = \iint \mathcal{D}\bar{q}\mathcal{D}q \exp\left\{-(\bar{q}S^{-1}q) + \frac{g^2}{2}\langle(\bar{q}\Gamma\mathcal{A}q)(\bar{q}\Gamma\mathcal{A}q)\rangle_D\right\},$$
$$Z_{\mathcal{A}\mathcal{A}} = \left\langle \exp\left\{-\frac{g}{2}(f\mathcal{A}\mathcal{A}F)\right\}\right\rangle_D, \qquad \langle(\bullet)\rangle_D \doteq \int \mathcal{D}\mathcal{A}\, e^{-\frac{1}{2}(\mathcal{A}D^{-1}\mathcal{A})}(\bullet). \quad (2)$$

Our first step is to transform these partition functions so that they could be rewritten in terms of meson and glueball fields. Below we briefly explain the procedure on the example of the quark-antiquark bound states defined by $Z_{q\bar{q}}$. Further details on our model can be found in [27,28].

First, we allocate the one-gluon exchange between colored biquark currents

$$L_2 = \frac{g^2}{2} \sum_{f_1 f_2} \iint dx_1 dx_2 \left( \bar{q}_{f_1}(x_1) i\gamma_\mu t^A q_{f_1}(x_1) \right) D_{\mu\nu}^{AB}(x_1, x_2) \left( \bar{q}_{f_2}(x_2) i\gamma_\nu t^B q_{f_2}(x_2) \right). \tag{3}$$

By isolating the color-singlet combination and performing a Fierz transformation we rewrite

$$L_2 = \frac{2g^2}{9} \sum_{Jf_1 f_2} C_J \iint dx dy \, \mathcal{J}_{Jf_1 f_2}(x, y) \, D(y) \, \mathcal{J}_{Jf_1 f_2}^\dagger(x, y), \tag{4}$$

where $\mathcal{J}_{Jf_1 f_2}(x, y) = \bar{q}_{f_1}(x + \xi_1 y) \, O_J \, q_{f_2}(x - \xi_2 y), O_J = \{I, i\gamma_5, i\gamma_\mu, \gamma_5\gamma_\mu, i[\gamma_\mu, \gamma_\nu]/2\}$, $x_1 = x + \xi_1 y$, $x_2 = x - \xi_2 y$, $\xi_i = m_{f_i}/(m_{f_1} + m_{f_2})$ and $C_J$ are the Fierz coefficients corresponding to the different spin combinations $O_J = \{I, i\gamma_5, i\gamma_\mu, \gamma_5\gamma_\mu, i[\gamma_\mu, \gamma_\nu]/2\}$.

Let us consider a system of orthonormalized basis functions $\{U_Q(x)\}$:

$$\int dx \, U_Q(x) U_{Q'}(x) = \delta_{QQ'}, \qquad \sum_Q U_Q(x) U_{Q'}(y) = \delta(x - y), \tag{5}$$

Particularly, it may read as:

$$U_{nl\{\mu\}}(x) \sim T_{l\{\mu\}}(x) \, L_n^{(l+1)}\left(2cx^2\right) e^{-cx^2}, \tag{6}$$

where $c > 0$ is a parameter, $T_{l\{\mu\}}$ is spherical harmonics and $L_n^{(l+1)}(x)$ are the Laguerre polynomials.

We expand the biquark nonlocal current on the orthonormalized basis as follows:

$$\begin{aligned} D(y) \, \mathcal{J}_{Jf_1 f_2}^\dagger(x, y) &= \sqrt{D(y)} \int dz \, \delta(z - y) \sqrt{D(z)} \, \mathcal{J}_{Jf_1 f_2}^\dagger(x, z) \\ &= \sum_Q \int dz \, \sqrt{D(y)} U_Q(y) \cdot \sqrt{D(z)} U_Q(z) \, \mathcal{J}_{Jf_1 f_2}^\dagger(x, z). \end{aligned} \tag{7}$$

Then, we define a vertex function $V_{QJ}(x, y)$

$$\bar{q}_{f_1}(x) \, V_{QJ}(x, y) \, q_{f_2}(x) \doteq \frac{2}{3} \sqrt{C_J} \sqrt{D(y)} U_Q(y) \bar{q}_{f_1}(x + \xi_1 y) \, O_J \, q_{f_2}(x - \xi_2 y) \tag{8}$$

and a colorless biquark current localized at the center of masses:

$$\mathcal{J}_{\mathcal{N}}(x) \doteq \int dy \, \bar{q}_{f_1}(x) \, V_{QJ}(x, y) \, q_{f_2}(x), \quad \mathcal{J}_{\mathcal{N}}^\dagger(x) = \mathcal{J}_{\mathcal{N}}(x), \quad \mathcal{N} = \{Q, J, f_1, f_2\}. \tag{9}$$

One can diagonalize $L_2$ on the basis $\{U_Q(x)\}$ and then, (4) takes a form:

$$L_2 = \frac{g^2}{2} \sum_{\mathcal{N}} \int dx \, \mathcal{J}_{\mathcal{N}}(x) \mathcal{J}_{\mathcal{N}}(x). \tag{10}$$

We use a Gaussian path-integral representation for the exponential

$$e^{\frac{g^2}{2}\sum_{\mathcal{N}}(\mathcal{J}_{\mathcal{N}}^2)} = \left\langle e^{g(B_{\mathcal{N}}\mathcal{J}_{\mathcal{N}})} \right\rangle_B, \qquad \langle(\bullet)\rangle_B \doteq \int \prod_N \mathcal{D}B_N \, e^{-\frac{1}{2}(B_{\mathcal{N}}^2)}(\bullet), \qquad \langle 1 \rangle_B = 1 \tag{11}$$

by introducing auxiliary meson fields $B_{\mathcal{N}}(x)$. Then, we obtain

$$Z_{q\bar{q}} = \left\langle \iint \mathcal{D}\bar{q}\mathcal{D}q \exp\left\{-(\bar{q}S^{-1}q) + g(B_{\mathcal{N}}\mathcal{J}_{\mathcal{N}})\right\} \right\rangle_B \tag{12}$$

that allows us to take explicitly the path integration over quark variables and obtain

$$Z_{q\bar{q}} \to Z_B = \langle \exp\left\{\mathrm{Tr}\ln\left[1 + g(B_{\mathcal{N}}V_{\mathcal{N}})S\right]\right\}\rangle_B, \tag{13}$$

where $\mathrm{Tr} \doteq \mathrm{Tr}_c\mathrm{Tr}_\gamma$; $\mathrm{Tr}_c$ and $\mathrm{Tr}_\gamma$ are traces taken on color and spinor indices, correspondingly.

We introduce a hadronization Ansatz and identify $B_{\mathcal{N}}(x)$ fields with mesons carrying quantum numbers $\mathcal{N}$.

Let us collect all quadratic field configurations ($\sim B_{\mathcal{N}}^2$) in the 'kinetic' term and isolate higher-order terms describing the interaction between mesons in $W_{res}[g\,B_{\mathcal{N}}] \sim 0(g^3\,B_{\mathcal{N}}^3)$.

Then, we obtain the path integral written in terms of meson fields $B_{\mathcal{N}}$ as follows:

$$Z_{q\bar{q}} \to Z_B = \int \prod_{\mathcal{N}} \mathcal{D}B_{\mathcal{N}} \, \exp\left\{-\frac{1}{2}\sum_{\mathcal{N}\mathcal{N}'}(B_{\mathcal{N}}\left[\delta^{\mathcal{N}\mathcal{N}'} - \alpha_s\lambda_{\mathcal{N}\mathcal{N}'}\right]B_{\mathcal{N}'}) + W_{res}[g\,B_{\mathcal{N}}]\right\}, \tag{14}$$

where the LO kernel of the meson polarization function $\lambda_{\mathcal{N}\mathcal{N}'}$ is defined and $\alpha_s \doteq g^2/4\pi$.

## 2.1. UV Regularization of Meson Field and Strong Charge

It is a difficult problem to describe a composite particle within QFT which operates with free fields quantized by imposing commutator relations between creation and annihilation operators. The asymptotic *in-* and *out-* states are constructed by means of these operators acting on the vacuum state. Physical processes are described by the elements of the S-matrix taken for the relevant *in-* and *out-* states. The original Lagrangian requires renormalization, i.e., the transition from unrenormalized quantities such as mass, wave function, and coupling constant to the physical or renormalized ones.

The appropriate diagonalization of the Fourier transform of the polarization kernel

$$\iint dxdy \, U_Q(x)\lambda_{\mathcal{N}\mathcal{N}'}(p,x,y) \, U_Q(y) = \delta^{\mathcal{N}\mathcal{N}'} \, \lambda_{\mathcal{N}}(-p^2),$$

$$\lambda_{\mathcal{N}}(-p^2) = \frac{8\,C_J}{9\pi^3} \int d^4k \, |V_J(k)|^2 \, \Pi_{\mathcal{N}}(k,p) \tag{15}$$

is equivalent to the solution of the corresponding ladder BSE. The kernel $\lambda_{\mathcal{N}\mathcal{N}'}(p,x,y)$ is real and symmetric ($p$ - the meson momentum), variational methods can be applied for its evaluation.

The Fourier transform of the vertex function $V_J(k)$ determined by the LO one-gluon exchange and the polarization (self-energy) kernel $\Pi_{\mathcal{N}}(p)$ of the meson in Equation (15) are defined as follows:

$$V_J(k) \doteq \int d^4x \, U_J(x)\sqrt{D(x)}\, e^{-ikx}, \tag{16}$$

$$\Pi_{\mathcal{N}}(k,p) \doteq -\frac{N_c}{4!}\,\mathrm{Tr}\left[O_J\tilde{S}_{m_1}\left(\hat{k} + \xi_1\hat{p}\right)O_{J'}\tilde{S}_{m_2}\left(\hat{k} - \xi_2\hat{p}\right)\right].$$

The graphical representation of the meson self-energy function $\lambda_{\mathcal{N}}(-p^2)$ is given in Figure 1.

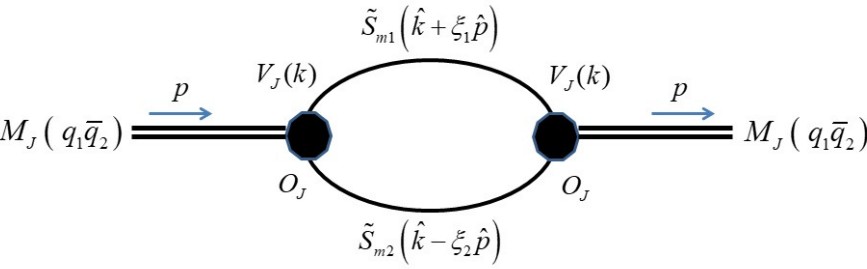

**Figure 1.** The graphical representation of the meson self-energy function $\lambda_{\mathcal{N}}(-p^2)$.

The gluon $\tilde{D}(p)$ (in Feynman gauge) and quark propagator $\tilde{S}_{m_1}(\hat{p})$ defined in Euclidean momentum space read:

$$\tilde{D}_{\mu\nu}^{AB}(p) = \delta^{AB}\delta_{\mu\nu} \cdot \tilde{D}_0(p), \qquad \tilde{D}_0(p) = \frac{1}{p^2} = \int_0^\infty ds\, e^{-sp^2},$$

$$\tilde{S}_{m_f}(\hat{p}) = \frac{1}{-i\hat{p} + m_f} = (i\hat{p} + m_f) \cdot \tilde{S}_{m_f}^0(p), \qquad \tilde{S}_{m_f}^0(p) = \int_0^\infty dt\, \exp[-t(p^2 + m_f^2)]. \tag{17}$$

In relativistic quantum-field theory a stable bound state of $n$ massive particles shows up as a pole in the S-matrix with a center of mass energy. Accordingly, we go into the meson mass shell $-p^2 = M_J^2$ and expand the quadratic term in Equation (14) as follows:

$$(B_{\mathcal{N}}[1 - \alpha_s\lambda_{\mathcal{N}}(-p^2)]B_{\mathcal{N}}) = (B_{\mathcal{N}}[1 - \alpha_s\lambda_{\mathcal{N}}(M_{\mathcal{N}}^2) - \alpha_s\dot{\lambda}_{\mathcal{N}}(M_{\mathcal{N}}^2)[p^2 + M_{\mathcal{N}}^2]B_{\mathcal{N}}). \tag{18}$$

Then, we rescale the boson field and strong charge as

$$B_{\mathcal{N}}(x) \rightarrow B_R(x)/\sqrt{\alpha_s\dot{\lambda}_{\mathcal{N}}(M_{\mathcal{N}}^2)}, \qquad g\,B_{\mathcal{N}}(x) \rightarrow g_R\,B_R(x), \qquad \dot{\lambda}_{\mathcal{N}}(z) \doteq \frac{d\lambda_{\mathcal{N}}(z)}{dz} \tag{19}$$

If we require a condition

$$1 - \alpha_s\lambda_{\mathcal{N}}(M_{\mathcal{N}}^2) = 0 \tag{20}$$

one obtains the Lagrangian of meson field $B_R$ with the mass $M$ and Green's function $(p^2 + M_{\mathcal{N}}^2)^{-1}$ in the fully renormalized partition function (the conventional form) as follows:

$$Z = \int \mathcal{D}B_R \exp\left\{-\frac{1}{2}\left(B_R\left[p^2 + M_{\mathcal{N}}^2\right]B_R\right) + W_{res}[g_R\,B_R]\right\}. \tag{21}$$

It is easy to find that regularizations in Equation (19) lead to another requirement:

$$Z_M = 1 - \alpha_R\dot{\lambda}_{\mathcal{N}}(M_{\mathcal{N}}^2) = 0, \tag{22}$$

that is nothing else but the 'compositeness' condition which means that the renormalization constant of the mesonic field $Z_M$ is equal to zero and bare meson fields are absent in the consideration.

Since the calculation of the Feynman diagrams proceeds in the Euclidean region where $k^2 = -k_E^2$, the vertex function $V_J(k)$ decreases rapidly for $k_E^2 \to \infty$ and thereby provides ultraviolet convergence in the evaluation of any diagram.

## 2.2. IR Regularization of the Green Functions

Ultraviolet singularities in the model have been removed by renormalizations of wave function and charge, but infrared divergences remain in Equation (20) because of propagators in Equation (17). The QCD vacuum structure remains unclear and the definition of the explicit quark and gluon propagator encounters difficulties in the confinement region. Particularly, IR behaviors of the quark and gluon propagators are not well-established and need to be more specified [26]. It is clear that conventional forms of the propagators in Equation (17) cannot adequately describe the hadronization dynamics and the currents and vertices used to describe the connection of quarks and gluons inside hadrons cannot be purely local. Presently, any widely accepted and rigorous analytic solutions to these propagators are still missing.

In our previous papers, specific forms of quark and gluon propagators were exploited [27,28]. These propagators were entire analytic functions in Euclidean space and represented simple and reasonable approximations to the explicit propagators calculated in the background of vacuum gluon field obtained in [4].

On the other hand, there are theoretical results predicting an IR behavior of the gluon propagator. Particularly, a gluon propagator was inversely proportional to the dynamical gluon mass [30] at the momentum origin $p^2 = 0$, while others equaled to zero [31,32]. Numerical lattice studies [33] and renormalization group analysis [34] also indicated an IR-finite behavior of gluon propagator.

Below we follow these theoretical predictions in favor of an IR-finite behavior of the gluon propagator and exploit a scheme of 'soft' infrared cutoffs on the limits of scale integrations for the scalar parts of both propagators as follows:

$$D_0(x) = \frac{1}{(2\pi)^2 x^2} \to D_\Lambda(x) = \frac{1}{(2\pi)^2} \int_{\Lambda^2}^{\infty} ds \, \exp[-s\,x^2] ,$$

$$\tilde{S}_{m_f}^0(p) \to \tilde{S}_{m_f}^\Lambda(p) = \int_0^{1/\Lambda^2} dt \, \exp[-t\,(p^2 + m_f^2)] . \tag{23}$$

Propagators $D_\Lambda(x)$ and $\tilde{S}_{m_f}^\Lambda(p)$ do not have any singularities in the finite $x^2$- and $p^2$- planes in Euclidean space, thus indicating the absence of a single gluon (quark) in the asymptotic space of states. The analytic confinement means the absence of real mass poles in the gluon and quark propagators. Particularly, the Fourier transformation of the gluon propagator in Equation (23) reads

$$\tilde{D}_\Lambda(p) = \left\{ 1 - \exp[-p^2/4\Lambda^2] \right\} / p^2 = 1/(4\Lambda^2) + p^2/(32\Lambda^4) + O(p^4) . \tag{24}$$

An IR parametrization is hidden in the energy-scale $\Lambda$ of confinement domain. The analytic confinement disappears as $\Lambda \to 0$. Note, propagators in Equation (23) differ from those used previously in [25,27–29] and represent lower bounds to the explicit ones.

*2.3. Meson Mass Equation*

The dependence of meson mass $M$ on $\alpha_s$ and other model parameters $\{\Lambda, m_1, m_2\}$ is defined by Equation (20). Furthermore, it is convenient to go to dimensionless co-ordinates, momenta and masses as follows:

$$x_\nu \cdot \Lambda \to x_\nu, \quad k_\nu/\Lambda \to k_\nu, \quad p_\nu/\Lambda \to p_\nu, \quad m_1/\Lambda \to \mu_1, \quad m_2/\Lambda \to \mu_2, \quad M/\Lambda \to \mu. \tag{25}$$

The polarization kernel $\lambda_{\mathcal{N}}(-p^2)$ in Equation (15) is natively obtained real and symmetric that allows us to find a simple variational solution to this problem. Choosing a trial Gaussian function for the ground-state mesons:

$$U(x,a) = \frac{2a}{\pi} \exp\left\{-ax^2\right\}, \quad \int d^4x \, |U(x,a)|^2 = 1, \quad a > 0 \tag{26}$$

we obtain a variational form of Equation (20) for meson masses as follows:

$$1 = \alpha_s \cdot \max_{a>0} \lambda(\mu, \mu_1, \mu_2, a) = \alpha_s \cdot \hat{\lambda}(\mu, \mu_1, \mu_2). \tag{27}$$

Further we exploit Equation (27) in different ways, by solving either for $\alpha_s$ at fixed masses $\{\mu, \mu_1, \mu_2\}$, or for $\mu$ by keeping $\alpha_s$ and $\{\mu_1, \mu_2\}$ fixed.

## 3. Effective Strong Coupling in the IR Region

Understanding of both high-energy and hadronic phenomena is necessary to know the strong coupling in the nonperturbative domain at low mass scale. Despite important results and constraints obtained from experiments, most investigations of the IR behavior of $\alpha_s$ have been theoretical, a number of approaches have been explored with their own benefits, justifications, and limitations.

The QCD coupling may feature an IR-finite behavior (e.g., in [35]). Particularly, the averaged IR value of strong coupling obtained from analyzing jet shape observables in $e^+e^-$ annihilation is finite and modest: $\langle \alpha_s \rangle = 0.47 \pm 0.07$ for the energy interval $E < 2$ GeV [36]. The stochastic vacuum model approach to high-energy scattering found that $\alpha_s \sim 0.81$ in the IR region [37]. Some theoretical arguments lead to a nontrivial IR freezing point, particularly, the analytical coupling freezes at the value of $4\pi/\beta_0$ within one-loop approximation [26]. The phenomenological evidence for $\alpha_s$ finite in the IR region is much more numerous [9,10,38].

There is an indication that the most fundamental Green's functions of QCD, such as the gluon and quark propagators may govern the detailed dynamics of the strong interaction and the effective strong charge [39]. Therefore, in the present paper we perform a new investigation of the IR behavior of $\alpha_s$ as a function of mass-scale $M$ by using the IR-confined propagators defined in Equation (23).

In our previous investigation, we studied the mass-scale dependence of $\alpha_s(M)$ within another realization of analytical confinement and determined it by fitting physical masses of mesons [28,29]. This strategy led to a smooth decreasing behavior of $\alpha_s(M)$, but the result was depending on a particular choice of model parameters, namely, the masses $m_1$ and $m_2$ of two constituent quarks composing a meson.

However, any physical observable, including $\alpha_s$, should not depend on the particular scheme of calculation, by definition. This kind of dependence is most pronounced in leading-order QCD and often used to test and specify uncertainties of theoretical calculations for physical observables. There is no common agreement of how to fix the choice of scheme.

Our idea is to investigate the behavior of the strong effective coupling $\alpha_s$ only in dependence of mass-scale $\mu$ by solving Equation (27). In doing so, the dependencies on $\mu_1$ and $\mu_2$ may be removed by revealing and substituting indirect dependencies of $\mu_i = \mu_i(\mu)$.

For this purpose, we analyze the meson masses estimated in [28,29] in dependence of fixed parameters $m_1$ and $m_2$, there. Then, one may easily notice a pattern: $m_1 + m_2 > M$ for light ($\pi$ and $K$) and $m_1 + m_2 < M$ for other mesons ($\rho$, $K^*$, ..., $\eta_b$, $\Upsilon$). Hereby, the constituent quark masses $\{m_{ud}, m_s, m_c, m_b\}$ were obtained by fitting $\alpha_s(M)$ at physical masses of $\{D^*, D_s^*, J/\Psi, \Upsilon\}$ and then, we calculate $(m_1 + m_2)/M = \{0.816, 0.824, 0.935, 0.992\}$ of these mesons, consequently.

A similar pattern is also revealed in the case of our earlier model with 'frozen' strong coupling not depending on mass scale [27]. Also, it was stressed that the self-energy function $\lambda(M, m_1, m_2)$ was low sensitive under significant changes of parameters $m_1$, $m_2$ (see Figure 2 in [27]).

Therefore, not losing the general pattern, we can substitute an 'average' dependence $m_1 = m_2 = M/2$. As mentioned above, this assumption is not able to change drastically the behavior of $\alpha_s$. Controversially, we now define $\alpha_s$ more self-consistently, in dependence only on the mass-scale $\mu = M/\Lambda$ by eliminating the direct presence of constituent quark masses.

Finally, we calculate a variational solution $\hat{\alpha}_s$ to $\alpha_s$ in dependence on a dimensionless energy-scale ratio $\mu$ as follows:

$$\hat{\alpha}_s(\mu) = 1/\hat{\lambda}(\mu, \mu/2, \mu/2). \tag{28}$$

The behavior of new variational upper bound $\hat{\alpha}_s(\mu)$ to $\alpha_s(\mu)$ is plotted in Figure 2. The slope of the curve depends on $\Lambda > 0$, but the value at origin remains fixed for any $\Lambda > 0$ and equals to

$$\hat{\alpha}_s(0) = \hat{\alpha}_s^0 = 1.03198, \qquad \text{or} \qquad \hat{\alpha}_s^0/\pi = 0.328489. \tag{29}$$

We use the meson mass $M$ as the appropriate characteristic parameter, so the coupling $\hat{\alpha}_s(M)$ is defined in a time-like domain ($s = M^2$). On the other hand, the most of known data on $\alpha_s(Q)$ is possible in space-like region [1]. The continuation of the invariant charge from the time-like to the space-like region (and vice versa) is elaborated by making use of the integral relationships (see, e.g., [40]). Particularly, there takes place a relation:

$$\alpha_s(q^2) = q^2 \int_0^\infty \frac{ds}{(s + q^2)^2} \hat{\alpha}_s(s) \tag{30}$$

A detailed study of this transformation deserves a separate consideration and below we just note that at origin ($q^2 = -s = 0$) both representations converge:

$$\alpha_s(0) = \hat{\alpha}_s(0) \int_0^\infty \frac{dt}{(1 + t)^2} = \hat{\alpha}_s(0) \cdot 1. \tag{31}$$

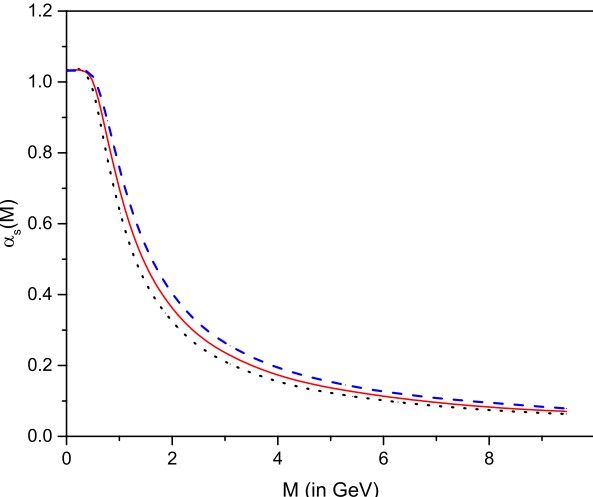

**Figure 2.** Mass-dependent effective strong coupling $\hat{\alpha}_s(M)$ for different values of confinement scale (dots for $\Lambda$ = 216 MeV, solid line for $\Lambda$ = 236 MeV and dashes for $\Lambda$ = 256 MeV.

Therefore, the obtained freezing value $\hat{\alpha}_s(0) = 1.03198$ may be compared with those obtained as continuation of $\alpha_s(Q)$ in space-like domain. Particularly, in the region below the $\tau$-lepton mass the strong coupling value is expected between $\alpha_s(M_\tau) \approx 0.34$ [1] and an IR fix point $\alpha_s(0) = 2.972$ [41]. Moreover, a use of $\overline{MS}$ renormalization scheme leads to value $\alpha_s(0) = 1.22 \pm 0.04 \pm 0.11 \pm 0.09$ for confinement scale $\Lambda_{QCD} = 0.34 \pm 0.02$ GeV [9].

We conclude that our IR freezing value $\hat{\alpha}_s^0/\pi = 0.32849$ does not contradict above mentioned predictions and it is in a reasonable agreement with other quoted estimates: $\alpha_s^0/\pi = 0.265$ [42] and $\alpha_s^0/\pi = 0.26$ [43] as well as other phenomenological predictions [44,45].

It should be stressed that despite some similar behaviors, the definition and origin of our mass-dependent coupling $\hat{\alpha}_s$ is quite distinct from the RG flow expected in QCD.

It is important to stress that we do not aim to obtain the behavior of the coupling constant at all scales. At moderate $M^2 = -p^2$ we obtain $\alpha_s$ in coincidence with the QCD predictions. However, at large mass-scale (above 10 GeV) $\hat{\alpha}_s$ decreases faster. The reason is the use of confined propagators in the form of entire functions in Equation (23). Then, the convolution of entire functions leads to a rapid decreasing in Euclidean (or, a rapid growth in Minkowskian) space of physical matrix elements once the mass and energy of the reaction have been fixed. Consequently, the numerical results become sensitive to changes of model parameters at large masses and energies.

Note, any physical observable must be independent of the particular scheme and mass by definition, but in (28) we obtain $\hat{\alpha}_s$ in dependence on scaled mass $M/\Lambda$. This kind of scale dependence is most pronounced in leading-order QCD and often used to test and specify uncertainties of theoretical calculations for physical observables. Conventionally, the central value of $\alpha_s(\mu)$ is determined or taken for $\mu$ equaling the typical energy of the underlying scattering reaction. There is no common agreement of how to fix the choice of scales.

Below, we will fix the model parameter $\Lambda$ by fitting the scalar glueball (two-gluon bound state) mass.

## 4. Lowest Glueball State

Most known experimental signatures for glueballs are an enhanced production in gluon-rich channels of radiative decays and some decay branching fractions incompatible with ($q\bar{q}$) states. Particularly, there are predictions expecting non-$q\bar{q}$ scalar objects, such as glueballs in the mass range $\sim 1.5 \div 1.8$ GeV [46,47].

Some references favor the $f_0(1710)$ and $f_0(1810)$ as the lightest glueballs [48], while heavy glueball-like states (pseudoscalar, tensor, ...) are expected in the mass range $M_G \sim 2.4 \div 4.9$ GeV [1].

Gluodynamics has been extensively investigated within quenched lattice QCD simulations. A use of fine isotropic lattices resulted in a value 1475 MeV for the scalar glueball mass [16]. An improved quenched lattice calculation at the infinite volume and continuum limits estimates the scalar glueball mass equal to $1710 \pm 50 \pm 80$ MeV [49].

Among different glueball models, the two-gluon bound states are the most studied purely gluonic systems in the literature, because when the spin-orbital interaction is ignored ($\ell = 0$), only scalar and tensor states are allowed. Particularly, the lightest glueballs with positive charge parity can be successfully modeled by a two-gluon system in which the constituent gluons are massless helicity-one particles [50].

Below we consider a pure two-gluon scalar bound state with $J^{PC} = 0^{++}$. By omitting details of intermediate calculations (similar to those represented in the previous section) we define the scalar glueball mass $M_{0^{++}}$ from equation:

$$1 - \frac{8\,\hat{\alpha}}{3\pi} \max_{a>0} \int dz\, e^{izp}\, \Pi_G(z) = 0\,, \qquad p^2 = -M_{0^{++}}^2\,, \tag{32}$$

where

$$\Pi_G(z) \doteq \iint dt\, ds\, U(t,a)\sqrt{W_\Lambda(t)}\, D_\Lambda\left(\frac{t+s}{2}+z\right) D_\Lambda\left(\frac{t+s}{2}-z\right) \sqrt{W_\Lambda(s)}\, U(s,a)$$

is the self-energy (polarization) function of the scalar glueball and $W_\Lambda(s)$ is a potential function connecting scalar gluon currents. The ground-state basis $U(t,a)$ may be chosen as in Equation (26). Then, we can estimate an upper bound to the scalar glueball mass by using the effective mass-dependent coupling defined in Equation (28).

Our model has a minimal set of free parameters: $\{\hat{\alpha}, \Lambda, m_{ud}, m_s, m_c, m_b\}$. The glueball mass depends on $\{\hat{\alpha}, \Lambda\}$. We fix $\Lambda$ by fitting the expected glueball mass. Particularly, for $\Lambda = 236$ MeV and $\hat{\alpha}(M_G)$ defined in Equation (28) we obtain new estimates:

$$M_{0^{++}} = 1739\,\text{MeV}\,, \qquad \hat{\alpha}(M_{0^{++}}) = 0.451\,. \tag{33}$$

The new value of $M_{0^{++}}$ in (33) agrees not only with our previous estimate [27], but also with other predictions expecting the lightest glueball located in the scalar channel in the mass range $\sim 1500 \div 1800$ MeV [12,16,46,51]. The often referred quenched QCD calculations predict $1750 \pm 50 \pm 80$ MeV for the mass of the lightest glueball [17]. The recent quenched lattice estimate with improved lattice spacing favors a scalar glueball mass $M_G = 1710 \pm 50 \pm 58$ MeV [49].

Another important property of the scalar glueball is its size, the 'radius' which should depend somehow on the glueball mass. We estimate the glueball radius roughly as follows:

$$r_{0^{++}} \sim \frac{1}{2\Lambda} \sqrt{\frac{\int d^4x\, x^2\, W_\Lambda(x)\, U^2(x)}{\int d^4x\, W_\Lambda(x)\, U^2(x)}} \approx \frac{1}{394.3\,\text{MeV}} \approx 0.51\,\text{fm}\,. \tag{34}$$

This may indicate that the dominant forces binding gluons are provided by vacuum fluctuations of correlation length $\sim 0.5$ fm. On the other side, typical energy-momentum transfers inside a scalar glueball should occur in the confinement domain $\sim 236$ MeV $\sim 0.85$ fm, rather than at the chiral symmetry breaking scale $\Lambda_\chi \sim 1$ GeV $\sim 0.2$ fm.

From (33) and (34) we deduce that

$$r_{0^{++}} \cdot M_{0^{++}} \approx 4.41 \,.$$

This value may be compared with the prediction ($r_G \cdot M_G = 4.16 \pm 0.15$) of quenched QCD calculations [17, 49].

The gluon condensate is a non-perturbative property of the QCD vacuum and may be partly responsible for giving masses to certain hadrons. The correlation function in QCD dictates the value of corresponding condensate. Particularly, with $\Lambda = 236$ MeV and $\hat{\alpha}_s = 0.451$ we calculate the lowest non-vanishing gluon condensate in the leading-order (ladder) approximation:

$$\frac{\hat{\alpha}_s}{\pi} \left\langle F^A_{\mu\nu} F_A^{\mu\nu} \right\rangle = \frac{16 N_c}{\pi} \Lambda^4 \approx 0.0214 \, \text{GeV}^4$$

which is in accordance with a refereed value [52]

$$\alpha_s \left\langle G^2 \right\rangle = (7.0 \pm 1.3) \cdot 10^{-2} \, GeV^4 \,, \qquad \text{or,} \qquad \frac{\alpha_s}{\pi} \left\langle G^2 \right\rangle = (2.2 \pm 0.4) \cdot 10^{-2} \, GeV^4 \,.$$

## 5. Meson Mass Spectrum

Below we consider the most established sector of hadron physics, the spectrum of conventional (pseudoscalar **P**($0^{-+}$) and vector **V**($1^{--}$)) mesons.

In previous investigations with analytic confinement [27–29], we fixed all the model parameters ($\Lambda, \hat{\alpha}_s, m_{ud}, m_s, m_c, m_b$) by fitting the real meson masses.

In the present paper, the universal confinement scale $\Lambda = 236 \, MeV$ is fixed by fitting the scalar glueball mass. In addition, the effective strong coupling $\hat{\alpha}_s$ is unambiguously determined by Equation (28).

Therefore, we derive meson mass formula Equation (20) by fitting the meson physical masses with adjustable parameters $\{m_{ud}, m_s, m_c, m_b\}$.

This results in a new final set of model parameters (in units of MeV) as follows:

$$\Lambda = 236 \,, \quad m_{ud} = 227.6 \,, \quad m_s = 420.1 \,, \quad m_c = 1521.6 \,, \quad m_b = 4757.2 \,. \tag{35}$$

The constituent quark mass values fall into the expected range. The present numerical least-squares fit for meson masses and the values for the model parameters supersede our previous results in [27,28] obtained by exploiting different types of analytic confinement and running coupling.

It is known that light pseudoscalar mesons ($\pi(140)$ and $K(494)$) are much lighter than their vector counterparts ($\rho(770)$ and $K^*(892)$). Any correct description of these light pseudoscalar mesons requires an indispensable implementation of dynamical chiral symmetry breaking in the working theoretical model. Therefore, no results for the pion and kaon masses are represented in Table 1.

**Table 1.** Estimated masses of conventional mesons $M_P$ and $M_V$ (in units of MeV) for model parameters (35) compared to the recent experimental data [1].

| $0^{-+}$ | $M_P$ | Data | $1^{--}$ | $M_V$ | Data |
|---|---|---|---|---|---|
| | | | $\rho$ | 774.3 | 775.26 |
| | | | $K^*$ | 892.9 | 891.66 |
| $D$ | 1893.6 | 1869.62 | $\Phi$ | 1010.3 | 1019.45 |
| $D_s$ | 2003.7 | 1968.50 | $D^*$ | 2003.8 | 2010.29 |
| $\eta_c$ | 3032.5 | 2983.70 | $D_s^*$ | 2084.1 | 2112.3 |
| $B$ | 5215.2 | 5259.26 | $J/\Psi$ | 3077.6 | 3096.92 |
| $B_s$ | 5323.6 | 5366.77 | $B^*$ | 5261.5 | 5325.2 |
| $B_c$ | 6297.0 | 6274.5 | $B_s^*$ | 5370.9 | 5415.8 |
| $\eta_b$ | 9512.5 | 9398.0 | $Y$ | 9526.4 | 9460.30 |

Note, we consider $\omega$ and $\Phi$ as 'pure' states without mixing. Also, we pass the $\eta - \eta'$ mixing, because this problem obviously deserves a separate and complicated consideration due to a possible gluon admixture to the conventional $q\bar{q}$-structure of the $\eta'$.

Our present model has only five free parameters ($\Lambda$ and four masses of constituent quarks) and a constraint self-consistent equation for $\alpha_s$. Nevertheless, our estimates on the conventional meson masses represented in Table 1 are in reasonable agreement with experimental data and the relative errors do not exceed 1.8 per cent in the whole range of mass scale.

## 6. Leptonic Decay Constants of Mesons

One of the important quantities in the hadron physics is the leptonic (weak) decay constant of meson. The precise knowledge of its value provides significant improvement in our understanding of various processes convolving meson decays. Particularly, the weak decay constants of light mesons are well established data and many collaboration groups have these with sufficient accuracy [1,53,54].

Therefore, the leptonic decay constant values (plotted in Figure 2 in dependence of meson physical mass) are often used to test various theoretical models.

A given meson in our model is characterized by its mass $M$, two of constituent quark masses $m_1$ and $m_2$ along the infrared confinement parameter $\Lambda$ universal for all hadrons, including exotic glueballs. The masses ($m_{ud}, m_s, m_c, m_b$) of four constituent quarks have been obtained by fitting the meson physical masses. Hereby, the effective strong coupling $\hat{\alpha}_s$ depends on the ratio $M/\Lambda$.

The leptonic decay constants which are known either from experiment or from lattice simulations is an additional characteristic of a given meson.

We define the leptonic decay constants of pseudoscalar and vector mesons as follows:

$$ip_\mu f_P = g_{ren} \frac{8N_c}{9} \int \frac{dk}{(2\pi)^4} V_P(k) \text{Tr} \left[ \gamma_\mu (1 - \gamma_5) \tilde{S} \left( \hat{k} + \xi_1 \hat{p} \right) i\gamma_5 \tilde{S} \left( \hat{k} - \xi_2 \hat{p} \right) \right],$$

$$\delta^{\mu\nu} M_V f_V = g_{ren} \frac{8N_c}{9} \int \frac{dk}{(2\pi)^4} V_V(k) \text{Tr} \left[ \gamma_\mu \tilde{S} \left( \hat{k} + \xi_1 \hat{p} \right) i\gamma_\nu \tilde{S} \left( \hat{k} - \xi_2 \hat{p} \right) \right], \tag{36}$$

where $g_{ren} = g/\sqrt{\alpha \dot{\lambda}(M_J)}$ is the renormalized strong charge and vertices $V_J(k)$ are defined in Equation (16).

The parameters $\Lambda, M_J, m_1, m_2$ and $\hat{\alpha}_s$ have been already fixed by considering the glueball and meson spectra, so these values in Equation (35) will be used to solve Equation (36) for $f_P$ and $f_V$.

In doing so we note a 'sawtooth'-type dependence of $f_J$ on meson masses (see Figure 3) that requires an additional parameterization to model more adequately this unsmooth behavior.

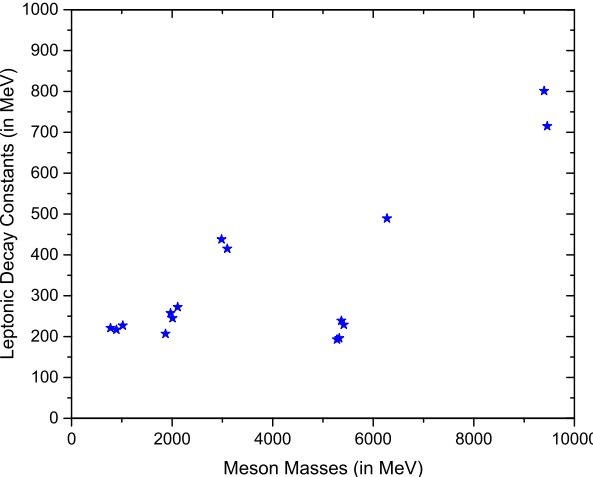

**Figure 3.** Experimental data on leptonic decay constants plotted versus physical masses of mesons.

For the meson mass Equation (27), the parameter $a$ in the basis function $U(x, a)$ served as a variational parameter to maximize the meson self-energy function $\lambda(M, m_1, m_2)$.

In contrast to this, for Equations (36) we introduce:

$$U(x, R_M) = \frac{R_M^2}{\pi \Lambda^2} \exp\left\{ -R_M^2 x^2 / 2 \right\}, \qquad R_M > 0, \tag{37}$$

where $R_M$ characterizes the 'size' of each meson $M$ in units of mass.

Then, we define the meson 'sizes' $R_M$ by solving Equation (36) with Equation (37) and fixed model parameters Equation (35).

Note, the 'size' parameters $R_M$ show the expected general pattern: the 'geometrical size' of a meson, which is proportional to $1/R_M$, shrinks when the meson mass increases.

The obtained values of meson 'sizes' and the best fit values estimated for the leptonic decay constants are represented in Table 2.

**Table 2.** Meson 'size' parameters $R_M$ (in units of GeV) and leptonic decay constants $f_P$ and $f_V$ (in units of MeV) compared to experimental data in [1,55–57].

| $0^{-+}$ | $R_M$ | $f_P$ | Data | Ref. | $1^{--}$ | $R_M$ | $f_V$ | Data | Ref. |
|---|---|---|---|---|---|---|---|---|---|
| | | | | | $\rho$ | 0.33 | 221 | $221 \pm 1$ | [1] |
| | | | | | $K^*$ | 0.38 | 217 | $217 \pm 7$ | [1] |
| $D$ | 0.93 | 207 | $206.7 \pm 8.9$ | [1] | $\Phi$ | 0.42 | 227 | $227 \pm 2$ | [1] |
| $D_s$ | 1.08 | 257 | $257.5 \pm 6.1$ | [1] | $D^*$ | 0.78 | 245 | $245 \pm 20$ | [57] |
| $\eta_c$ | 1.83 | 238 | $238 \pm 8$ | [56] | $D_s^*$ | 0.90 | 271 | $272 \pm 26$ | [57] |
| $B$ | 1.73 | 193 | $192.8 \pm 9.9$ | [55] | $J/\Psi$ | 2.40 | 416 | $415 \pm 7$ | [1] |
| $B_s$ | 2.18 | 239 | $238.8 \pm 9.5$ | [55] | $B^*$ | 3.34 | 196 | $196 \pm 44$ | [57] |
| $B_c$ | 3.34 | 488 | $489 \pm 5$ | [56] | $B_s^*$ | 0.92 | 228 | $229 \pm 46$ | [57] |
| $\eta_b$ | 3.80 | 800 | $801 \pm 9$ | [56] | $\Upsilon$ | 2.80 | 715 | $715 \pm 5$ | [1] |

## 7. Conclusions

In conclusion, we demonstrate that many properties of the low-energy phenomena such as strong running coupling, hadronization processes, mass generation for quark-antiquark and di-gluon bound

states may be explained reasonably within a QCD-inspired model with infrared-confined propagators. We derived a meson mass equation and by exploiting it revealed a specific new behavior of the strong coupling $\hat{\alpha}_s(M)$ in dependence of mass scale. An infrared freezing point $\hat{\alpha}_s(0) = 1.03198$ at origin $M = 0$ has been found and it did not depend on the particular choice of the confinement scale $\Lambda > 0$. A new estimate of the lowest (scalar) glueball mass has been performed and it was found at $\sim 1739$ MeV. The scalar glueball 'size' has also been calculated: $r_G \sim 0.51$ fm. A nontrivial value of the gluon condensate has also been obtained. We have estimated the spectrum of conventional mesons by introducing a minimal set of parameters: four masses of constituent quarks $\{u = d, s, c, b\}$ and $\Lambda$. The obtained values fit the latest experimental data with relative errors less than 1.8 percent. Accurate estimates of the leptonic decay constants of pseudoscalar and vector mesons have also been performed.

In the present paper, we considered neither quark-antiquark potential which may occur consistent with confinement, nor the role of the chiral condensate in the model, these questions deserve separate studies.

Note, the suggested model in its simple form is far from real QCD. However, our guess about the structure of the quark-gluon interaction in the confinement region, implemented by means of confined propagators and nonlocal vertices has been probed and the obtained numerical results were in reasonable agreement with experimental data in different sectors of low-energy particle physics. Since the model is probed and the parameters are fixed, the consideration may be extended to actual problems in hadron physics, such as spectra of other mesons (scalar, iso-scalar), higher glueball states, exotic states ($q\bar{q} + gg$ admixtures, tetraquark, X(3872) and Z(4430), ...), baryon decays ($\Lambda_b \rightarrow \Lambda^* + J/\Psi$) etc.

**Funding:** This research received no external funding.

**Conflicts of Interest:** The author declares no conflict of interest.

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
