# Peer review of "Strong Effective Coupling, Meson Ground States, and Glueball within Analytic Confinement"

_2571-712X, doi:10.3390/particles2020013_

Round 1
Reviewer 1 Report
The manuscript presents new results of the mass dependence of the strong running coupling, the mass of the lowest glueball and the pseudoscalar and vector quark-antiquark meson spectrum from a QCD-inspired model with 5 parameters. The obtained values are in good agreement with the experimental data.
I believe that the obtained results are interesting enough to warrant publication. However, I have a few points to be addressed by the author:
(1) It is stated by the author that
“One of the puzzles of hadron physics is the origin of the hadron masses. The Standard Model and, in particular, QCD operate only with fundamental particles (quarks, leptons, neutrinos), gauge bosons and the Higgs. It is not yet clear how to explain the appearance of the multitude of observed hadrons and elucidate the generation of their masses. Physicists have proposed a number of models that advocate different mechanism of the origin of mass from the most fundamental laws of physics. The calculation of the hadron mass spectrum in a quality comparable to the precision of experimental data remains actual.”
I do not fully agree with the author. Dynamical chiral symmetry breaking is the widely accepted mechanism responsible for the mass generation, see, for instance, arXiv:1811.01003v2, for a recent calculation of the dynamical quark mass function, which establishes the connection between the current quark masses (fom the Higgs mechanism) and the constituent masses of quarks inside hadrons.
(2) A question related to the above point: No results for the pion and kaon masses are presented. How could such a model ever describe the pion correctly, for which the implementation of dynamical chiral symmetry breaking is indispensable?
(3) I assume that the author means with analytic confinement the absence of real mass poles in the gluon and quark propagators. For the glueball calculation the constituent gluon mass is assumed to be zero. Is a zero gluon mass not in contraction with analytic confinement, in the sense that there is a (real) gluon mass pole at p^2=0? In any case, I am not really too worried about real mass poles of constituent quarks or gluons (see, for instance, arXiv:hep-ph/9911319v1 or arXiv:1707.09303 where quarks can be on-shell and are still confined in meson). I am more worried that “analytical” confinement is not reconcilable with a zero gluon mass.
(4) It is stated that the model gives a solution for mesons that is equivalent to the solution from the Bethe-Salpeter equation in ladder truncation. If this is the case, to which quark-antiquark interaction kernel in the BSE would the model then correspond to?
Author Response
Thank you very much for reviewing my manuscript.
I almost fully agree with remarks and criticisms made by the journal Rewiewers and
have revised the manuscript according to their comments. Hereby, revisions made in
the revised LaTeX source file are clearly highlighted (in yellow) in the corresponding
PDF output.
My response to Reviewer 1 is attached.
Kind regards,
Gurjav Ganbold

Reviewer 2 Report
The paper describes an improved version of a relativistic quark model for hadrons proposed by the same author some years ago. The improvement regards the ansatz for the gluon and quark propagators. Although the paper does not bring a new insight into the mechanisms of confinement and chiral symmetry breaking, the results seem to be correct and phenomenologically appealing.
Therefore, I recommend the paper for publication after the author addresses the following points:
- In section 2, a LO result for the quark-antiquark and di-gluon bound states is presented. Since apparently the approach at that point is perturbative, it would be important to describe the Feynman diagrams considered. Adding some figures could be useful.
- The way the meson fields appear in eq. (4) is far from obvious. Although the author cites some previous references, it would be important to explain briefly the procedure followed. In particular, it would interesting to elaborate on the relation between mesons and quark currents.
- Also in section 2, a renormalization procedure for the meson field and strong charge is presented. The author could explain more clearly the form of the divergence and the counter-terms required. The way it is presented looks just as a Taylor expansion.
- In section 3 a result for the strong coupling as a function of the meson mass is presented. Considering the meson mass as a continuous parameter does not sound appropriate since we know that it should be generated dynamically in QCD. The way the results are presented suggest some flow for the coupling similar to the one expected from RG equations. A clear distinction between RG flow in QCD and the model should be made.
- Finally, it would be interesting if the author describes: i) how the model would lead to a quark-antiquark potential consistent with confinement, ii) the role of the chiral condensate in the model.
Author Response
Thank you very much for reviewing my manuscript.
I almost fully agree with remarks and criticisms made by the journal Rewiewers and
have revised the manuscript according to their comments. Hereby, revisions made in
the revised LaTeX source file are clearly highlighted (in yellow) in the corresponding
PDF output.
My response to Reviewer 2 is attached.
Kind regards,
Gurjav Ganbold

Round 2
Reviewer 1 Report
The author has addressed and answered my points and concerns and he has changed the manuscript accordingly. I believe the manuscript is now ready for publication.
Reviewer 2 Report
The author has successfully addressed the points I raised. The revised manuscript is ready for publication.